# Revealing Hydrogen States in Carbon Structures by Analyzing the Thermal Desorption Spectra

**Yury S. Nechaev** [1,*], **Evgeny A. Denisov** [2], **Nadezhda A. Shurygina** [1], **Alisa O. Cheretaeva** [3], **Ekaterina K. Kostikova** [4], **Sergei Yu. Davydov** [5] **and Andreas Öchsner** [6]

[1] G.V. Kurdjumov Centre of Metals Science and Physics, I.P. Bardin Research Institute for Ferrous Metallurgy, Radio Str., 23/9, Build. 2, 105005 Moscow, Russia; shnadya@yandex.ru

[2] Solid State Electronics Department, St. Petersburg State University, Universitetskaya nab. 7/9, 198904 St. Petersburg, Russia; denisov70@bk.ru

[3] Research Institute of Progressive Technologies, Togliatti State University, Belorusskaya Str. 14, 445020 Togliatti, Russia; a.cheretaeva@tltsu.ru

[4] Institute of Applied Mathematical Research, Karelian Research Centre of the Russian Academy of Science, Pushkinskaya Str., 11, 185910 Petrozavodsk, Russia; fedorova@krc.karelia.ru

[5] Ioffe Physical Technical Institute, RAS, Polytechnicheskaya Str., 26, 194021 St. Petersburg, Russia; sergei_davydov@mail.ru

[6] Faculty of Mechanical and Systems Engineering, Esslingen University of Applied Sciences, Kanalstrasse, 33, 73728 Esslingen, Germany; andreas.oechsner@gmail.com

* Correspondence: yuri1939@inbox.ru; Tel.: +7-495-491-0262

**Abstract:** An effective methodology for the detailed analysis of thermal desorption spectra (TDS) of hydrogen in carbon structures at micro- and nanoscale was further developed and applied for a number of TDS data of one heating rate, in particular, for graphite materials irradiated with atomic hydrogen. The technique is based on a preliminary description of hydrogen desorption spectra by symmetric Gaussians with their special processing in the approximation of the first- and the second-order reactions. As a result, the activation energies and the pre-exponential factors of the rate constants of the hydrogen desorption processes are determined, analyzed and interpreted. Some final verification of the results was completed using methods of numerical simulation of thermal desorption peaks (non-Gaussians) corresponding to the first- and the second-order reactions. The main research finding of this work is a further refinement and/or disclosure of poorly studied characteristics and physics of various states of hydrogen in microscale graphite structures after irradiation with atomic hydrogen, and comparison with the related results for nanoscale carbon structures. This is important for understanding the behavior and relationship of hydrogen in a number of cases of high energy carbon-based materials and nanomaterials.

**Keywords:** graphite irradiated with atomic hydrogen; thermal desorption spectra; rate constants; activation energies; processes physics

## 1. Introduction

Carbon materials and nanomaterials, especially those based on graphene, are widely used as energetic materials associated with electrochemical conversion and energy storage in fuel cells, supercapacitors, and lithium-ion batteries [1,2]. Carbon materials, including isotropic graphite, were recently revised for being used as a plasma-facing material of fusion reactors [3]. In addition, they still may be the ideal materials for diverter plates [4]. At the same time, the behavior and states of hydrogen in such systems have been poorly studied, despite its possible effect on technological processes and physicochemical properties. Therefore, it seems expedient to study in more detail the states and characteristics of hydrogen in such materials using the known thermal desorption spectroscopy data.

In this regard, it should be noted the presence of a large amount of experimental data on the thermal desorption spectra (TDS) of hydrogen for various carbon materials

and nanomaterials [4–6] with different hydrogen content, in particular, for graphite after irradiation with atomic hydrogen [7–12]. Most of these TDS data have not yet been sufficiently processed or analyzed in detail. Additionally, as noted in many of these studies, as well as in some others, for example [13–15], there are difficulties with the approximation and interpretation of TDS data—the major gaps within the existing knowledge in this area. It should also be added that in some of the studies noted above, the approximation of second-order reactions was used, and in others, the approximation of first-order reactions.

In the present study, all these aspects have been taken into account. Hence, the present study objectives/procedures include the following points formulated below.

Firstly, a further development of the methodology [16–21] and its applications, relevant to TDS data [7–12], will be performed. The characteristics and atomic mechanisms of processes of thermal desorption of hydrogen from such materials [7–12] will be determined and analyzed, on the basis of using the results, methods and approaches given in [3–28].

Secondly, it should be noted that, in the present study, the approximations of both the first-order and second-order reactions will be used and compared, taking into account the physical aspects.

Thirdly, it should be also emphasized that, in this study, the main attention will be paid to the disclosure of the poorly studied physics of the desorption processes, and not to the thorough mathematical description of the thermal desorption spectra, prevailing in most studies noted above.

## 2. Methodology and Materials

Further development of the methodology [16–21] for "processing" and detailed analysis of hydrogen thermal desorption spectra for carbon materials and nanomaterials, for cases of one single heating rate, has been completed and used in the present study in relation to experimental data [7–12] for graphite subjected to irradiation with atomic hydrogen. The developed technique [20] is not less informative, but much less time-consuming in experimental terms compared to the generally accepted Kissinger method [13–15], which demands the use of several heating rates, and has strict limits of applicability [4,19,20].

The methodology [20] contains several successive steps of its implementation, including the use of several plausibility checks and some final verification of the results, with the help of numerical modeling methods [21].

The first step consists of the standard deconvolution of the studied thermal desorption spectrum by the smallest number of symmetrical Gaussians (peaks) corresponding to different temperatures ($T_{max}$) of the maximum desorption rate.

The second stage consists of determining (in the approximation of a first-order reaction) for each of the above noted Gaussians, from the temperature dependence of the desorption flux ($-\mathrm{d}\theta/\mathrm{d}t = J_H$) divided by the heating rate ($\beta$), since upon heating at a constant rate $\mathrm{d}T = \beta \mathrm{d}t$, the rate constants ($K(T)$) of hydrogen desorption at different temperatures (around $T_{max}$), and hence the activation energy ($Q$), as well as the pre-exponential factor ($K_0$) of the hydrogen rate are constants.

For such estimates, the formal kinetics equation for the first-order reactions was used:

$$-\left(\frac{1}{\beta}\right)\frac{\mathrm{d}\theta}{\mathrm{d}t} = -\frac{\mathrm{d}\theta}{\mathrm{d}T} = K\frac{\theta}{\beta} = K_0\left(\frac{\theta}{\beta}\right)\exp\left(-\frac{Q}{RT}\right) \tag{1}$$

where $t$ is the time, $T$ is the temperature, $R$ is the universal gas constant, $\theta = (C/C_0)$ is the relative average concentration of hydrogen in the carbon sample (relevance for the considered Gaussian, for the given $T$ and $t$), $\theta = 1$ at $t = 0$.

Then, the quantity $Q^*$ (related to the quantity $Q$) was evaluated, by using the corresponding expression for the first-order reactions, (see Equation (2)) and the values of $T_{max}$ and $K(T_{max})$ for the Gaussian under consideration. The proximity of the obtained values of $Q$ and $Q^*$ is one of the plausibility checks.

Such an expression can be obtained from the condition of the maximum desorption rate ($\{d(J_H/\beta)/dT = 0\}$ and/or $\{d^2\theta/dT^2 = 0\}$), as follows:

$$Q^* \approx \frac{RT_{max}^2 K(T_{max})}{\beta} \tag{2}$$

where quantities $T_{max}$ and $K(T_{max})$ can be taken (in a satisfactory approximation) from the above obtained results for the considered Gaussian.

From the above shown Equations (1) and (2), it follows that the $Q^*$ quantity can be also determined by the Kissinger method (i.e., from the linear dependence of $\ln(T_{max}^2/\beta)$ versus $1/T_{max}$), relevant for first-order desorption processes. However, as shown in [4,19,20], this is valid only in the absence of close neighboring thermal desorption peaks (near the considered one).

In any case, it is also advisable to estimate the value of $K(T_{max})$ for a sufficiently self-manifested (at $T_{max}$) thermal desorption peak, using the following expression for the first-order processes:

$$K(T_{max}) = -\left(\frac{1}{\theta_{max}}\right)\left(\frac{d\theta}{dt}\right)_{T_{max}} \tag{3}$$

In which the value of $\theta_{max}$ can first be taken about of 0.5 (with an error not more 15%), it can be refined from a numerical analysis [21] of the corresponding TDS data; the quantities $T_{max}$ and $(d\theta/dt)_{Tmax}$ can be approximated as those for the considered Gaussian. Hence, the quantity $Q^*$ can be re-evaluated by using Equation (2).

The next stage consists of determining in the approximation of a second-order reaction for each of the above noted Gaussians, from the temperature dependence of the desorption flux ($-d\theta/dt = J_H$), divided by the heating rate ($\beta$), the rate constants ($K(T)$) of hydrogen desorption at different temperatures (about $T_{max}$), and hence, the quantities $Q$ and $K_0$.

For such estimates, the formal kinetics equation for the second-order reactions was used:

$$-\left(\frac{1}{\beta}\right)\frac{d\theta}{dt} = -\frac{d\theta}{dT} = K\frac{\theta^2}{\beta} = K_0\left(\frac{\theta^2}{\beta}\right)\exp\left(-\frac{Q}{RT}\right) \tag{4}$$

The related quantity $Q^*$ was evaluated, by using the expression obtained from the condition of $\{d^2\theta/dT^2 = 0\}$, as follows:

$$Q^* \approx \frac{2RT_{max}^2\theta(T_{max})K(T_{max})}{\beta} \tag{5}$$

where the value of $\theta(T_{max})$ can first be taken of about 0.5 (with an error not more 15%), it can be refined from a numerical analysis [21] of the TDS data; quantities $T_{max}$ and $K(T_{max})$ can be taken (in a satisfactory approximation) from the above obtained results for the considered Gaussian.

From the above shown Equations (4) and (5), it follows that the $Q^*$ quantity can be also evaluated by the Kissinger method (i.e., from the linear dependence of $\ln(T_{max}^2/\beta)$ versus $1/T_{max}$), relevance for the second-order desorption processes.

The value of $K(T_{max})$ for the considered self-manifested (at $T_{max}$) peak can be estimated, using the following expression for the second-order processes:

$$K(T_{max}) = -\left(\frac{1}{\theta_{max}}\right)^2\left(\frac{d\theta}{dt}\right)_{T_{max}} \tag{6}$$

In which the quantity $\theta_{max}$ can first be taken of about 0.5, and then refined for the considered spectrum (from a numerical analysis [21] of related TDS data), the quantities $T_{max}$ and $(d\theta/dt)_{Tmax}$ can be approximated as those for the considered Gaussian. Hence, the quantity $Q^*$ can be re-evaluated by using Equation (5).

The final stage, used in the important cases, is some verification of the above obtained results with the help of the numerical simulation [21] of the TDS data, in the approxi-

mation of reactions of both the first- and second-orders, along with taking into account Equations (2), (3), (5) and (6) and the obtained values of $Q^*$ and $K(T_{max})$. It should be noted that in this case the considered TDS spectra are approximated not by symmetric Gaussians ($\theta_{max} = 0.5$), but non-symmetric peaks corresponding to processes of the first-order (usually, $\theta_{max} < 0.5$) or the second-order ($\theta_{max} > 0.5$, see Table A2).

Then, the physics and atomic mechanisms of desorption processes can be revealed through thermodynamic analysis of the obtained peak characteristics and comparison with the corresponding independent experimental and theoretical data.

The main goal of this methodology is to further disclose the not enough-studied characteristics and physics of various states of hydrogen in carbon materials and nanomaterials, especially in graphite after irradiation with atomic hydrogen [7–12], rather than a detailed mathematical description of the spectra. For this case, both the large difference and the large spread (scatter) of the known experimental and theoretical values of the thermodynamic characteristics of desorption processes are taken into account.

Finally, the real cases to support this methodology can be given, as follows:

(1)    The results of studying the thermal desorption of hydrogen (of different content) in some carbon nanostructures and graphite, particularly, in the graphane-like structures, see Ref. [20];

(2)    The results of studying the characteristics and physics of processes of thermal desorption of deuterium from isotropic graphite at 700–1700 K, see Ref. [19];

(3)    The results of the kinetic analysis of the hydrogen thermal desorption spectra for graphite and advanced carbon nanomaterials, see Ref. [18].

It should be also emphasized that the other existing methods (see, for instance, in this work together with in Refs. [18–20]) regarding hydrogen for various carbon materials and nanomaterial cannot, by themselves, provide a solution of the present study objectives.

## 3. Results

### 3.1. Results of Processing the TDS Data for Pyrolytic Graphite after Irradiation with Atomic Hydrogen

In [7] (Denisov and Kompaniets), the method of the temperature desorption spectroscopy (TDS) was used for studying the atomic hydrogen interaction with pyrolytic graphite of a density of $\rho = 2.186$ g cm$^{-3}$. On the microscale, it had a layered structure with a spacing of 0.5–1.0 μm in the growth (*c*-axis) direction. Graphite samples were made in the form of ribbons measuring $1 \times 40 \times 0.5$ mm. The surface of the ribbon was parallel to the basal graphite plane. The sample was attached to current leads and placed into a vacuum chamber. The residual pressure of mainly hydrogen was kept at $10^{-8}$ torr. The temperature, which can be varied according to a specified law, was measured with a W/WRe thermocouple. The desorbed hydrogen was detected by a magnetic sectorial mass spectrometer. Prior to sorption experiments, the samples were annealed for a long time at 1473 K. At the end of annealing, the temperature was momentarily raised to 1673 K. Purified hydrogen was supplied to the chamber through a diffusion filter. Atomization was carried out when the gas passed near a 100-μm-diameter tungsten filament heated to 2773 K. The filament was arranged parallel to the sample at a distance of 5–8 mm so that the atomic hydrogen flux struck the surface parallel to the basal plane of graphite. The irradiation dose was calculated with regard for the inlet hydrogen pressure, atomization yield, and mutual arrangement of the sample and the atomizer. The probability of hydrogen atomization on tungsten heated to 2373 K was taken equal to 0.3. During the irradiation exposure (at $T_{irr.} = 873$ K, for $t_{irr.} = 7.5 \times 10^3$ s), the hydrogen pressure was $10^{-2}$ torr. The flux of hydrogen atoms toward the front side of the sample was estimated at $5 \times 10^{13}$ cm$^{-2}$ s$^{-1}$ in view of the experiment geometry. During the TDS study (with a linear heating up to 1673 K), it was recorded the total release of $H_2$ as $2.8 \times 10^{14}$, from the above noted sample of volume of 0.02 cm$^3$ and mass of 0.044 g, containing about $2 \times 10^{21}$ carbon atoms. Hence, the average (in relation to the whole sample) hydrogen atomic fraction is as $(H/C)_{\Sigma vol.} \sim 2.5 \times 10^{-7}$. Furthermore, in relation to the sample monolayer surface, i.e., if all the released hydrogen were located

only at the sample-free surface, a reasonable quantity can be evaluated as $(H/C)_{\Sigma surf.}$ ~0.4, which corresponds to the coverage (in terminology [10]) of about 40%.

To estimate the sticking coefficient of hydrogen atoms on the surface of graphite, it is necessary to use the data for hydrogen uptake obtained at low doses of hydrogen atoms, while the reemission flux from the surface is negligible. So, at the dose of atoms of $3 \times 10^{15}$ $(H^0)$ cm$^{-2}$ (60 s irradiation), the hydrogen uptake is $6.5 \times 10^{14}$ (H) cm$^{-2}$, that yields a sticking coefficient of about ~0.22.

Results of deconvolution and processing the experimental data [7] are presented in Figure 1a,b, and Tables A1 and A2 (all Tables from this work are presented in the Appendix A).

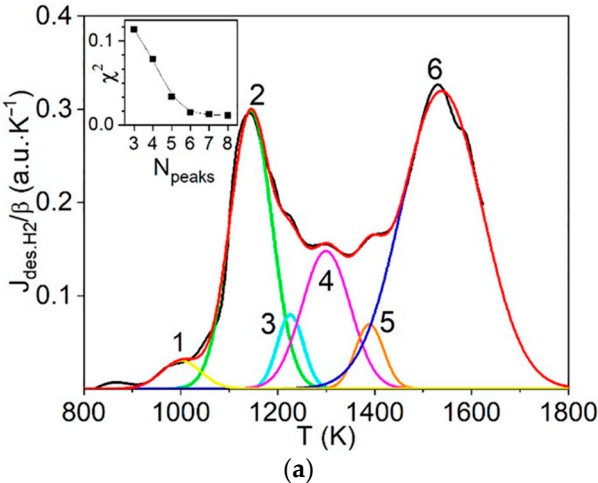
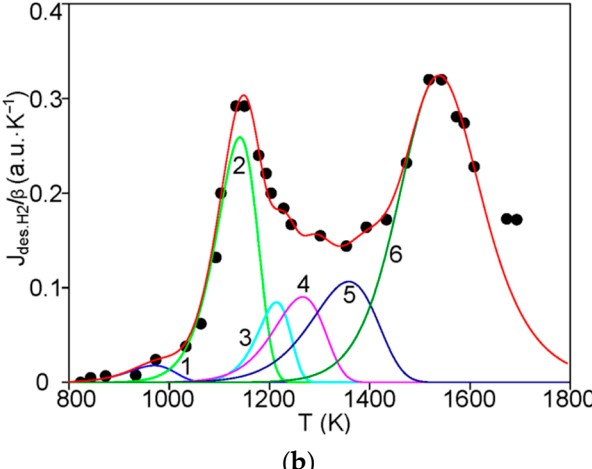

(a)  (b)

**Figure 1.** Processing of the thermal desorption spectrum (heating rate β = 25 K s$^{-1}$, see Figure 1 in [7]) of hydrogen for pyrolytic graphite (tape with sizes 0.5 × 1 × 40 mm) subjected irradiation (at $T_{irr.}$ = 873 K, for time $t_{ir}$r. = 7.5 × 10$^3$ s) with atomic hydrogen (flux ~5 × 10$^{13}$ cm$^{-2}$ s$^{-1}$), by using a W-wire (at about 2773 K) as atomizer, as follows: (**a**) Deconvolution, with the help of the methodology [20], by Gaussians (peaks ## 1–6). The dependence of the quadratic parameter of the theoretical curve from the experimental one on the Gaussian number is shown in the inset. Hence, it follows a suitable number of peaks (as 6). (**b**) Deconvolution, with the help of the numerical simulation [21], by five non-Gaussians (peaks ## 1–5) corresponding to the first-order reactions and by one non-Gaussian (peak #6) corresponding to the second-order reaction.

Approximation of the thermal desorption spectrum from Figure 1a by five, seven and eight Gaussians (see the inset in Figure 1a) does not lead to significant changes in the characteristics of the main peaks. Therefore, these results are not presented here.

The results of the numerical simulation of these six peaks as non-Gaussians (Table A2) are satisfactorily (with the physics revealing, especially for the main peaks) consistent with the results of processing of the six Gaussians (Table A1).

### 3.2. Interpretation of Peak #2 and #4

The obtained quantities, in the first-order approximation, for peak #2 (one of two main peaks) in Figure 1a (Gaussian: $T_{max\#2}$ = 1145 K, γ = 0.25, $Q_{\#2(1\_ord.)G.}$ ≈ 201 kJ mol$^{-1}$ and $K_{0\#2(1\_ord.)G.}$ ≈ 7.3 × 10$^8$ s$^{-1}$ (Table A1)) and in Figure 1b (non-Gaussian: $T_{max\#2}$ = 1122 K, γ = 0.22, $Q_{\#2(1\_ord.)}$ ≈ 230 kJ mol$^{-1}$ and $K_{0\#2(1\_ord.)}$ ≈ 2.8 × 10$^{10}$ s$^{-1}$ (Table A2)) are, as it is shown below, suitable for diffusion processes in carbon lattice, noted in [16,17] as processes III, accompanying with the reversible trapping [13–15,26] of the diffusant (hydrogen atoms) by some chemisorption "centers". It can be used models "F*" and/or "F", described in [16,22], corresponding to chemisorption of H atom on the graphite basal sites. For such processes, the apparent diffusion activation energy is as $Q_{III[1,6]}$ ≈ 250 kJ mol$^{-1}$, and the pre-exponential factor of the apparent diffusion coefficient is as $D_{0III[1,6]}$ ≈ 3 × 10$^{-3}$ cm$^2$ s$^{-1}$;

the physics of such processes has not been sufficiently studied up to the present (as is noted in [17] (access source via the Internet)).

Hence, the diffusion characteristic size ($L$) for peak #2 can be evaluated [16,17] as $L_{\text{Peak#2}} \approx (D_{0\text{III}[16]} / K_{0\#2(1\_\text{ord.})})^{1/2} \approx 3 \times 10^{-7}$ cm, that is close (within the errors) to the crystalline size [5] of $L_a \approx 1 \times 10^{-6}$ cm for graphite materials. This quantity can be also evaluated [16,17] as $L_{\text{Peak#2}} \approx (D_{\text{III}[16]}(T_{\text{max#2}})/K_{\#2(1\_\text{ord.})}(T_{\text{max#2}}))^{1/2} \approx 1 \times 10^{-7}$ cm. The latter quantity is consistent with the characteristic diffusion size ($L_{\text{irr.}}$) for the process under the temperature-time regime ($T_{\text{irr.}} \approx 873$ K, $t_{\text{irr.}} \approx 7.5 \times 10^{3}$ s) of irradiation of the sample with atomic hydrogen, which can be evaluated as $L_{\text{irr.}} \approx (D_{\text{III}[16]}(T_{\text{irr.}})\cdot t_{\text{irr.}})^{1/2} \approx 1.6 \times 10^{-7}$ cm.

In this connection, it is important to note that these quantities ($Q_{\#2(1\_\text{ord.})}$ and $K_{0\#2(1\_\text{ord.})}$, Table A2) for peak #2 in Figure 1b are satisfactorily (within the errors) consistent with the results of the Kissinger processing of spectra for the first self-manifesting peak (related to the considered peak #2) in graphite [7] for three heating rates ($\beta = 10$ K s$^{-1}$, $\beta = 25$ K s$^{-1}$ and $\beta = 100$ K s$^{-1}$).

Furthermore, it is expedient to take into account that the other obtained quantities, in the second-order approximation, for peak #2 in Figure 1a (Gaussian: $Q_{\#2(2\_\text{ord.})\text{G.}} \approx 402$ kJ mol$^{-1}$ and $K_{0\#2(2\_\text{ord.})\text{G.}} \approx 2 \times 10^{18}$ s$^{-1}$ (Table A1)) and in Figure 1b (non-Gaussian: $Q_{\#2(2\_\text{ord.})} \approx 340$ kJ mol$^{-1}$ and $K_{0\#2(2\_\text{ord.})} \approx 4 \times 10^{15}$ s$^{-1}$ (Table A2)) are not suitable enough for processes of recombination of hydrogen atoms in molecules and their desorption from chemisorption hydrogen "traps" on the sample free surface, i.e., for the case of the second-order Polanyi–Wigner equation [5].

Thus, the detailed analysis above shows that the chemisorption process, corresponding to peak #2 in Figure 1a,b can proceed as a first-order reaction with characteristics presented in Tables A1 and A2. It is highly likely that it can be rate-limited by the chemisorption diffusion process (with the reversible trapping by the chemisorption "centers" (model "F*"and/or "F" described in [16,22]). Within such an approach, the quantity $Q_{\#2(1\_\text{ord.})}$ refers to 1 mole of H.

A similar interpretation is possible for peak #4 in Figure 1a,b with rather close characteristics ($Q$ and $K_0$ (Tables A1 and A2)).

*3.3. Interpretation of Peaks #3 and #5*

The obtained quantities, in the first-order approximation, for the small peak #3 in Figure 1a (Gaussian: $T_{\text{max#3}} = 1226$ K, $\gamma = 0.04$, $Q_{\#3(1\_\text{ord.})\text{G.}} \approx 373$ kJ mol$^{-1}$ and $K_{0\#3(1\_\text{ord.})\text{G.}} \approx 5.7 \times 10^{15}$ s$^{-1}$ (Table A1)) and in Figure 1b (non-Gaussian: $T_{\text{max#3}} = 1198$ K, $\gamma = 0.05$, $Q_{\#3(1\_\text{ord.})} \approx 371$ kJ mol$^{-1}$ and $K_{0\#3(1\_\text{ord.})} \approx 1.2 \times 10^{16}$ s$^{-1}$ (Table A2)) are, as it is shown below, suitable for diffusion processes in carbon lattice, noted in [16,17] as processes IV, accompanying with the reversible trapping of the diffusant (hydrogen atoms) by some chemisorption "centers". For such processes, the apparent diffusion activation energy is as $Q_{\text{IV}[16]} = 365\pm50$ kJ mol$^{-1}$, and the pre-exponential factor of the apparent diffusion coefficient is as $D_{0\text{IV}[16]} \approx 6 \times 10^{2}$ cm$^2$ s$^{-1}$. The physics of such processes is described in [16,17], with using models "C" and/or "D" from [22] corresponding to chemisorption of H atoms on the graphite edge sites.

Hence, the diffusion characteristic size ($L$) for peak #3 can be evaluated as $L_{\text{Peak#3}} \approx (D_{0\text{IV}[16]} / K_{0\#3(1\_\text{ord.})})^{1/2} \approx 2 \times 10^{-7}$ cm or as $L_{\text{Peak#3}} \approx (D_{\text{IV}[16]}(T_{\text{max#3}}) / K_{\#3(1\_\text{ord.})}(T_{\text{max#3}}))^{1/2} \approx 3 \times 10^{-7}$ cm.

Furthermore, the other obtained quantities, in the second-order approximation, for peak #3 in Figure 1a (Gaussian: $Q_{\#3(2\_\text{ord.})\text{G.}} \approx 745$ kJ mol$^{-1}$ and $K_{0\#3(2\_\text{ord.})\text{G.}} \approx 8.6 \times 10^{31}$ s$^{-1}$ (Table A1)) and in Figure 1b (non-Gaussian: $Q_{\#3(2\_\text{ord.})} \approx 745$ kJ mol$^{-1}$ and $K_{0\#3(2\_\text{ord.})} \approx 2.0 \times 10^{32}$ s$^{-1}$ (Table A2)) are not suitable for processes of recombination of hydrogen atoms [5].

Thus, the analysis above shows that the process, corresponding to peak #3 in Figure 1a,b can proceed as a first-order reaction with characteristics presented in Tables A1 and A2. It is highly likely that it can be rate-limited by the diffusion with the reversible trapping of

the diffusant (H atoms) by the chemisorption "centers" (models "C" and/or "D" described in [16,22]). Within such an approach, the quantity $Q_{\#3(1\_ord.)}$ refers to 1 mole of H.

A similar interpretation is possible and for the small peak #5 in Figure 1a,b with rather close characteristics (Tables A1 and A2).

*3.4. Interpretation of Peak #6*

The obtained quantities, in the second-order approximation, for peak #6 (the second of two main peaks) in Figure 1a (Gaussian: $Q_{\#6(2\_ord.)G.} \approx 377$ kJ mol$^{-1}$ and $K_{0\#6(2\_ord.)G.} \approx 3.0 \times 10^{12}$ s$^{-1}$ (Table A1)) and in Figure 1b (non-Gaussian: $Q_{\#6(2\_ord.)} \approx 320$ kJ mol$^{-1}$ and $K_{0\#6(2\_ord.)} \approx 3.9 \times 10^{10}$ s$^{-1}$ (Table A2)) are, as it shows below, reasonable for processes of recombination of hydrogen atoms in molecules and their desorption from chemisorption hydrogen "traps" on the sample free surface (i.e., for the case of the second-order Polanyi–Wigner equation [5]). In this case, models "G" (corresponding to chemisorption of 2 H atoms per carbon atom on the graphite edge sites) and/or model "F" (corresponding to chemisorption of 2 H atoms per 2 carbon atoms on the graphite basal sites, those are described in [16,22]), may be suitable. Within such an approach, the quantity $Q_{\#6(2\_ord.)}$ refers to 1 mole of 2H.

It is important to add that the Kissinger processing for the second-order processes of the second self-manifesting peak (for $\beta = 10$ K s$^{-1}$ and $\beta = 25$ K s$^{-1}$), corresponding to peak #6 in Figure 1a,b results in quantities ($Q$ and $K_0$), which are close (within the errors) to the similar quantities ($Q_{\#6(2\_ord.)}$ and $K_{0\#6(2\_ord.)}$) for peak #6 in Table A2.

*3.5. Results of Processing the TDS Data for the (0001) Graphite Surface after Irradiation with Hydrogen Atoms, Relevance to Their Clustering*

In [10], metastable structures and recombination pathways for atomic hydrogen on the (0001) surface of highly oriented pyrolytic graphite (HOPG) were studied, by using the scanning tunneling microscopy (STM) and the temperature desorption spectroscopy (TDS), along with the density functional theory (DFT) calculations. It was stressed in [10] that in spite of increased theoretical activity aimed at determining the mobility of hydrogen atoms and the binding energy of hydrogen states on graphite, the recombination pathways for hydrogen atom adsorbates on graphite surfaces were still undetermined.

In [10], particularly, a STM image of the graphite surface after a 1 min dose of D atoms at 2200 K onto a room temperature HOPG sample was studied. A number of bright protrusions were observed in the image (the coverage was approximately 1%), which were identified [10] as clusters of chemisorbed deuterium atoms, as they only appeared after D dosing. The clusters presence was correlated [10] with the $D_2$ desorption peaks observed in the TPD spectra in [10].

It was noted in [10] that in accordance with earlier observations, no well-ordered superstructure of the adsorbed deuterium atoms was observed. Two different characteristic structures, labeled dimer A and dimer B, were observed to be dominant, with the dimer B structure as the most numerous. Dimers A were slim elongated spheroids, while dimer B structures were more rectangular in shape. Figure 2a in [10] (see Figure 2 in the present paper) shows a $D_2$ temperature programed desorption (TPD) spectrum from the HOPG surface after a 2 min D atom dose. The spectrum showed the double peak structure discussed in [10].

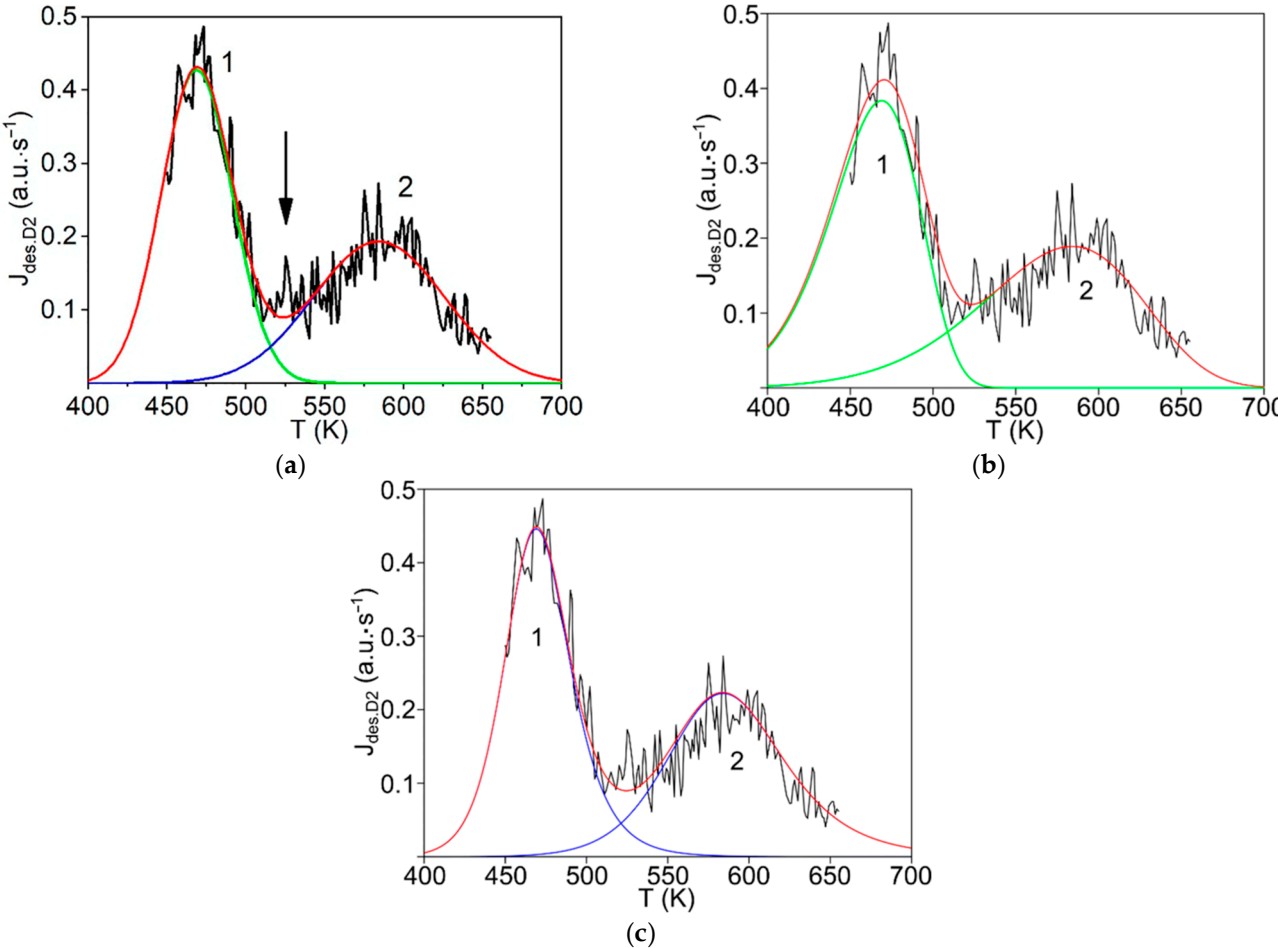

**Figure 2.** A mass 4 amu, i.e., $D_2$, thermal desorption spectrum [10] from the HOPG surface after a 2 min D atom dose at 2200 K onto a room temperature sample, i.e., $T_{irr.} \approx 300$ K and $t_{irr.} = 1.2 \times 10^2$ s (ramp rate: $\beta = 2$ K s$^{-1}$ below 450 K, $\beta = 1$ K s$^{-1}$ above), as follows: (**a**) Deconvolution, with the help of the methodology [20], by Gaussians (peaks ## 1,2). The arrow indicates the maximum temperature of the thermal anneal performed before recording the STM image of "dimer structures" of hydrogen atoms on the graphite surface. (**b**) Deconvolution, with the help of the numerical simulation [21], by non-Gaussians (peaks ## 1,2) corresponding to the first-order reactions. (**c**) Deconvolution, with the help of the numerical simulation [21], by non-Gaussians (peaks ## 1,2) corresponding to the second-order reactions.

In order to investigate if the peaks corresponded to different hydrogen adsorbate structures, it was performed a thermal annealing process to a temperature between the two peaks [10]. The arrow in Figure 2a in [10] (see Figure 2a in the present paper) indicated the maximum temperature (525 K) of the thermal annealing performed before recording the STM image in [10]. After the 525 K annealing, the sample was cooled to room temperature before being placed in the STM. It was found a significant decrease in the total coverage on the annealed sample, and now only dimer A structures were observed. Repeated experiments showed that annealing to temperatures of 500–600 K starting from a low coverage of D atoms led to a surface where dimer A was the dominant structure.

Hence, the experiment indicated that dimer A structures were stable against thermal annealing to 525 K, whereas dimer B structures were not [10]. Since no D atoms were visible on the surface after anneals to 600 K, i.e., above the high temperature peak in the TPD spectrum, the interpretation was that dimer B structures contributed to the first peak in the desorption spectra, while dimer A structures are associated with the second peak [10].

Based on the Polanyi–Wigner equation for first-order desorption and assuming a pre-exponential factor ($K_{0(1\_ord.)[10]}$) of $10^{13}$ s$^{-1}$, authors [10] have obtained that the two peaks at $T_{max\#1[10,8]}$ = 490 K (445 K) and $T_{max\#2[10,8]}$ = 580 K (560 K), observed in TPD spectra for deuterium (hydrogen) for graphite in works [8,10], correspond to the desorption energies $Q_{\#1(1\_ord.)[10,8]}$ of 133 kJ mol$^{-1}$ (121 kJ mol$^{-1}$) and $Q_{\#2(1\_ord.)[10,8]}$ 158 kJ mol$^{-1}$ (152 kJ mol$^{-1}$), respectively.

Our results of approximation and processing the TDS data [10] are presented in Figure 2a,b,c and Table A3. The obtained characteristics of the peaks #1 and #2 in Figure 2a,b for the first-order desorption processes (Table A3) considerably differ from ones [10] noted above, which were compared by authors [10] with results of their related DFT calculations to reveal the physics of these two dimer states.

Results of processing of the two non-Gaussians (peaks ## 1,2 in Figure 2b,c) in the approximation of the first- and second-order reactions are rather close to results in Table A3.

There are reasons, including the results of thermodynamic analysis [17,23] of a number of experimental data ([10] and others), to assume that the theoretical model of the dimer "A" and "B" structures in [10] is inadequate. This issue is discussed in the next Section.

### 3.6. Interpretation of Results of Processing the TDS Data for the (0001) Graphite Surface after Irradiation with Hydrogen Atoms, Relevance to Their Clustering

First of all, it is necessary to reveal the physics of these two dimer states [10] by taking into account the experimental data [11], as well as the results of thermodynamic analysis [17,23] of a number of related experimental data, including data [10]. In such a way, one can reveal that the two dimer structures are related to some surface nanoblisters (or nanoclusters), which contain the intercalated gaseous molecular hydrogen at a high pressure (it is at the expense of the association energy of the hydrogen atoms captured there in molecules, which has been described in [17,23]). These blisters are obviously localized at the subgrain boundaries. Indeed, in the STM images in [10], one can imagine something similar to subgrain-boundary network (with the subgrain size of about 2–5 nm) decorated by the bright "nanoprotrusions" (nanoclusters) mentioned in [10].

Taking into account all aspects noted above, one can suppose that dimer "A" (more stable) structures are related to surface nanoclusters (nanoblisters) localized mainly at some intermediate parts of the subgrain boundary regions. Furthermore, one can also suppose that dimer "B" structures are related to surface nanoblisters localized mainly at triple junctions (nodes) of the subgrain-boundary network in the HOPG samples. Such a model allows one to describe satisfactorily the desorption processes, corresponding to peaks ## 1, 2 in Figure 2, as it is shown below.

The quantities obtained by us (in the approximation of the first-order reactions) for peaks #1 and #2 in Figure 2a,b ($Q_{\#1.1\_ord.} \approx$ 64.5 kJ mol$^{-1}$, $K_{0\#1.1\_ord.} \approx 5.5 \times 10^5$ s$^{-1}$, $Q_{\#2.1\_ord.} \approx$ 54.6 kJ mol$^{-1}$ and $K_{0\#2.1\_ord.} \approx 1.5 \times 10^3$ s$^{-1}$ (Table A3)) can be related to the diffusion processes of type I and/or type II in intergranular and/or interfacial nanoregions of graphite materials, accompanied with the reversible chemisorption trapping of the diffusant ($H_2$), those are described in [16,17].

The characteristics of processes I and II are as $Q_{I[16]} \approx$ 20 kJ mol$^{-1}$ ($H_2$), $D_{0I[16]} \approx 3 \times 10^{-3}$ cm$^2$ s$^{-1}$, $Q_{II[16]} \approx$ 120 kJ mol$^{-1}$ ($H_2$), $D_{0II[16]} \approx 2 \times 10^3$ cm$^2$ s$^{-1}$. The physics of such processes is described in [16,17,26]. It was used the trapping models "G", "F" and "H" from [22], corresponding to chemisorption of 2 H atoms per 1 or 2 carbon atoms on the graphite basal or edge sites. Such diffusion processes of hydrogen molecules can proceed along the "short-circuited light diffusion paths" (considered, for instance in [2,16]) as grain boundaries and subgrain boundaries to the nearest free surface of the sample.

Hence, the desorption processes related to peaks #1 and #2 in Figure 2a,b can be related to such a diffusion of hydrogen molecule from the nanoblisters (i.e., from "A" and/or "B" "dimer" structures) along the neighboring grain boundaries and/or subgrain boundaries to the nearest crystalline fragment boundary. The crystalline fragment size can be of about

3–8 µm, as was shown in [8]. Within such models, the quantities $Q_{\#1.1\_ord.}$ and $Q_{\#2.1\_ord.}$ refer to 1 mole of $H_2$.

In this connection, it should be noted that the quantities ($Q_{\#1.2\_ord.}$, $K_{0\#1.2\_ord.}$, $Q_{\#2.2\_ord.}$ and $K_{0\#2.2\_ord.}$ (Table A3)) obtained for peaks ## 1,2 in Figure 2a,c in the approximation of the second-order reactions are, it seems, not suitable for processes of recombination of hydrogen atoms in molecules and desorption from chemisorption hydrogen "traps" on the sample free surface, mainly due to relatively low values of $Q_{\#1.2\_ord.}$ and $Q_{\#2.2\_ord.}$, which do not correspond to the related theoretical ones [10,22].

### 3.7. Results of Processing the TDS Data for the (0001) Graphite Surface after Irradiation with Hydrogen Atoms, Relevance to Their Adsorption

In [9], the adsorption of H and D atoms on HOPG surfaces (samples with a diameter of 9 mm and thickness of 1 mm) was studied with thermal desorption spectroscopy (TDS, along with electronic (ELS) and high-resolution electron-energy-loss (HREELS) spectroscopies. SEM pictures of the samples displayed crystalline graphite islands (fragments) and related surface "terraces" with about 3–8 µm diameter tilted against each other [9].

Thermal desorption spectra (see Figure 3) measured after admission of increasing D fluences to clean graphite at 150 K were studied in [9]. It was noted that up to a fluence of about 0.6 ML (1 ML = $3.8 \times 10^{15}$ cm$^{-2}$) the spectra exhibited a main peak at 490 K and a small satellite peak at 580 K. The main peak shifted to 500 K and developed a shoulder at 540 K above 0.6 ML D exposure. The amount of D desorbed as a function of applied D fluence as well as the D sticking coefficient were calculated [9] from the sample surface area and pumping speed of the differentially pumped quadrupole enclosure. The saturation coverage obtained from the spectra was found [9] as about 0.35, and the sticking coefficient was found between 0.25 and 0.5.

It was noted [9] that the flat "pedestal" seen in the spectra (see Figure 3) could be absent or replaced by a small peak at 300 K. The authors [9] believed that the origin of these features was probably water produced by an atom source after longer operation periods. In this connection, it should be noted that, according to the present study results, the origin of these features is related to $H_2$ physisorption [16,24].

The authors [9] believed that the reproducible main desorption peak at 500 K was assigned to desorption from the surface "terraces", which were observed on the SEM pictures, and its satellite peak at 580 K to desorption from imperfections on these "terraces". The not reproducible desorption features below 350 K were interpreted [9] as due to "islands" edge effects.

It was also noted [9] that the shape of the desorption spectra, both for deuterium (see Figure 3 below) and for protium exhibited for small D (or H) coverages the characteristics of first-order desorption, as follows: constant peak temperature and asymmetric. A first-order reaction analysis [9,25] of the leading edges of these spectra, with supposing the desorption rate of $D_2$ (or $H_2$) proportional to exp(-$Q$/RT), revealed an activation energy for desorption of deuterium as $Q_{\text{des.1\_ord.D2}[9]} \approx 92$ kJ mol$^{-1}$, and for desorption of hydrogen as $Q_{\text{des.1\_ord.H2}[9]} \approx 58$ kJ mol$^{-1}$.

It should be noted that these quantities ($Q_{\text{des.1\_ord.D2}[9]}$ and $Q_{\text{des.1\_ord.H2}[9]}$) are satisfactorily (within the errors) consistent with some of our results of processing of TDS data [9], relevance to the main peaks (see the next Section).

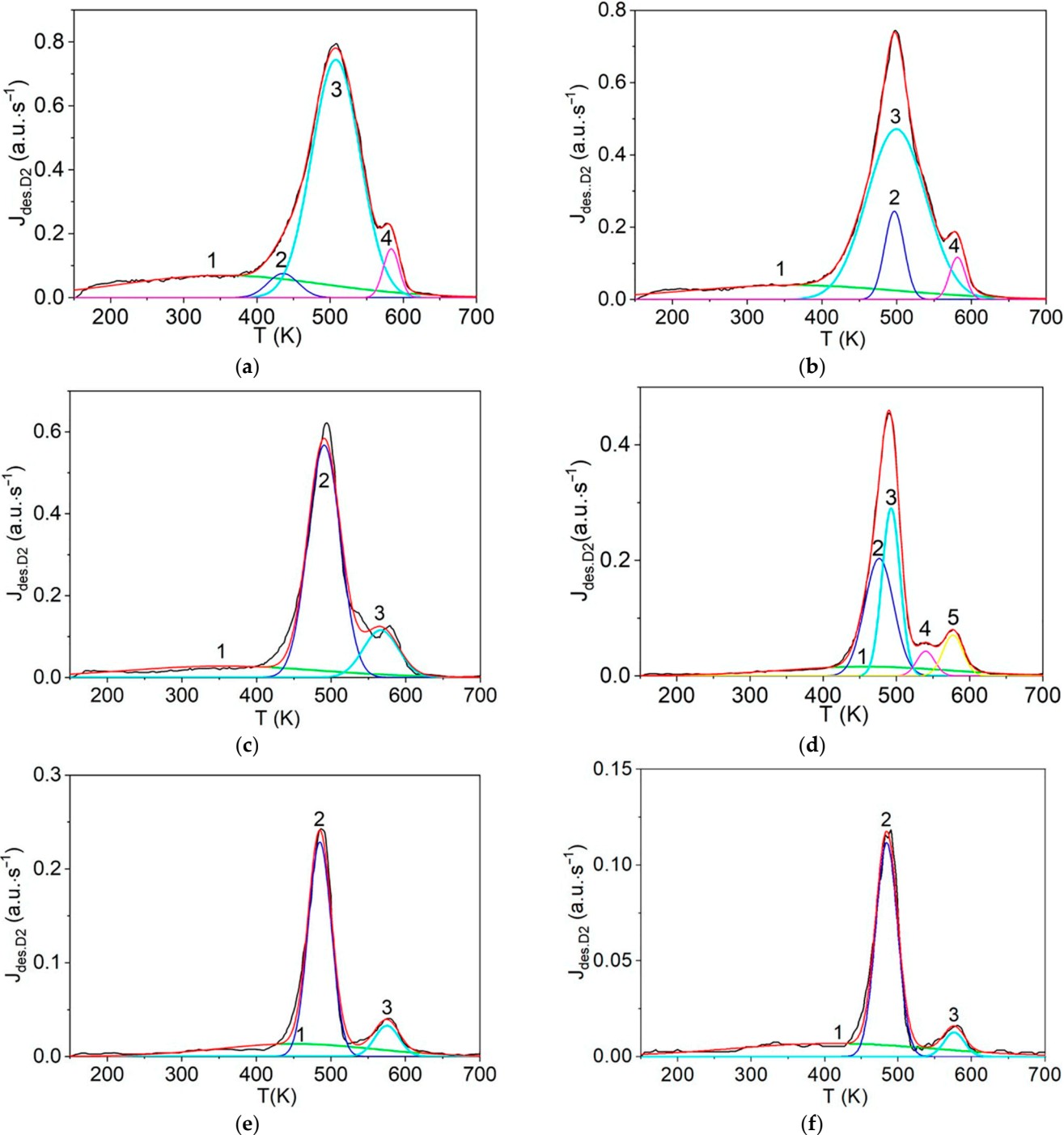

**Figure 3.** Approximation of the thermal desorption spectrum (from Figure 6 in [9]) measured after admitting D atoms to clean HOPG surfaces at 150 K: (**a**) By four Gaussians (exposure 5.8 ML (1 ML = 3.8 × 10$^{15}$ cm$^{-2}$)). (**b**) By four Gaussians (exposure 2.3 ML). (**c**) By three Gaussians (exposure 1.2 ML). (**d**) By five Gaussians (exposure 0.6 ML). (**e**) By three Gaussians (exposure 0.3 ML). (**f**) By three Gaussians (exposure 0.1 ML).

It was supposed [9] that D or H atoms, adsorbed on the surface terraces of graphite, desorbed recombinatively (the main peak). In this connection, it was noted [9] that normally, recombinative desorption of D (or H) atoms to gaseous $D_2$ (or $H_2$) molecules was described by a second-order rate law. Additionally, the first-order kinetics in the recombinative desorption process of adsorbed D (or H) atoms could indicate (according to [9]) that in the microscopic desorption dynamics the two recombining atoms did not move along the same

pathway on the potential energy surface. One atom played the role of an "activator" and dragged the other atom into the desorption event (according to [9]).

Our results of approximation and processing the experimental data [9], along with their interpretation are presented below (particularly, in Figures 3–5 and Tables A4–A6). In many respects, they differ significantly from those presented in [9].

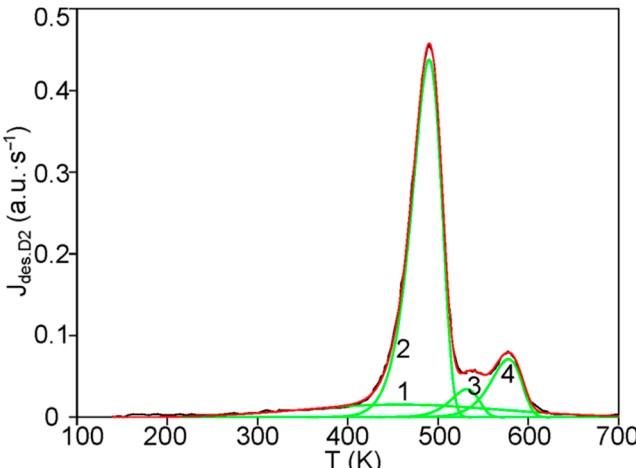

**Figure 4.** Deconvolution of the thermal desorption spectrum (from Figure 3d, Table A4) by non-Gaussians (peaks ## 1–4), with the help of the numerical simulation [19,20], in the approximation of the first-order reactions.

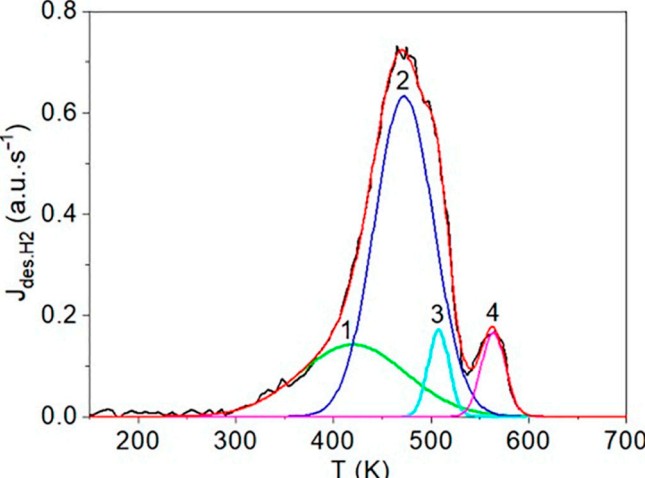

**Figure 5.** Deconvolution by four Gaussians (peaks #1–4) of the thermal desorption spectrum (from Figure 6 in [9]) measured after admitting H atoms (exposure 5.8 ML, 1 ML = $3.8 \times 10^{15}$ cm$^{-2}$) to clean HOPG surfaces at 150 K.

In terms of the physics, the results of the numerical simulation of the peaks in Figure 4 (Table A5) are satisfactorily consistent with the results of processing of the main Gaussians in Figure 3d (Table A4); the spread (and the related error) in values of the $Q$ and ln $K_0$ quantities is up to 9%.

### 3.8. Interpretation of Peak #1

The obtained quantities, in the first-order approximation, for peak #1 in Figure 3a ($T_{\mathrm{max\#1}}$ = 353 K, $\gamma$ = 0.24, $Q_{\#1(1\_ord.)} \approx$ 6.2 kJ mol$^{-1}$ and $K_{0\#1(1\_ord.)} \approx 4.9 \times 10^{-2}$ s$^{-1}$), in Figure 3b ($T_{\mathrm{max\#1}}$ = 361 K, $\gamma$ = 0.19, $Q_{\#1(1\_ord.)} \approx$ 6.3 kJ mol$^{-1}$ and $K_{0\#1(1\_ord.)} \approx 4.7 \times 10^{-2}$ s$^{-1}$), and in Figure 3c ($T_{\mathrm{max\#1}}$ = 354 K, $\gamma$ = 0.18, $Q_{\#1(1\_ord.)} \approx$ 6.5 kJ mol$^{-1}$ and $K_{0\#1(1\_ord.)}$

$\approx 5.6 \times 10^{-2}$ s$^{-1}$ (Table A4)) are rather suitable for the physisorption of H$_2$ on related "centers" on the graphite (0001) surface. In this connection, see related quantities and references in [10,16], along with the recent theoretical results [24]. Obviously, it is the case of the first-order Polanyi–Wigner equation [16], and the obtained quantity $Q_{\#1(1\_ord.)}$ refers to 1 mole of H$_2$ (see [24]).

### 3.9. Interpretation of Peaks #2 and #3

The obtained quantities (Table A4), in the first-order approximation, for peak #3 ($T_{max\#3}$ = 508 K, $\gamma$ = 0.67, $Q_{\#3(1\_ord.)} \approx 52.0$ kJ mol$^{-1}$ and $K_{0\#3(1\_ord.)} \approx 5.4 \times 10^3$ s$^{-1}$) and for peak #2 ($T_{max\#2}$ = 436 K, $\gamma$ = 0.04, $Q_{\#2(1\_ord.)} \approx 60.8$ kJ mol$^{-1}$ and $K_{0\#2(1\_ord.)} \approx 7.4 \times 10^5$ s$^{-1}$) in Figure 3a can be interpreted as the similar quantities (Table A3) for peaks #2 and #1 in Figure 2a,b (see Section 3.6). This is the case of the chemisorption diffusion processes of the first-order (with respect to the average concentration of the diffusant [13–16]), and the obtained quantities ($Q_{\#3(1\_ord.)}$ and $Q_{\#2(1\_ord.)}$) refer to 1 mole of H$_2$.

### 3.10. Interpretation of Peak #4

The obtained quantities, in the first-order approximation, for peak #4 in Figure 3a ($T_{max\#4}$ = 584 K, $\gamma$ = 0.05, $Q_{\#4\_3a(1\_ord.)} \approx 208$ kJ mol$^{-1}$ and $K_{0\#4\_3a(1\_ord.)} \approx 3.0 \times 10^{17}$ s$^{-1}$ (Table A5)) and for peak #4 in Figure 3b ($T_{max\#4}$ = 581 K, $\gamma$ = 0.04, $Q_{\#4\_3b(1\_ord.)} \approx 212$ kJ mol$^{-1}$ and $K_{0\#4\_3b(1\_ord.)} \approx 8.6 \times 10^{17}$ s$^{-1}$ (Table A4)) can be interpreted as the similar quantities for peak #4 in Figure 4a,b in [20] (for hydrogenated epitaxial single and few-layer graphene). It is the case of the first-order Polanyi–Wigner equation, and the obtained quantities ($Q_{\#4\_3a(1\_ord.)}$ and $Q_{\#4\_3b(1\_ord.)}$) refer to 1 mole of H.

### 3.11. Results of Processing the TDS Data for Hydrogen Adsorption on Terraces and Terrace Edges of Graphite (0001) Surface after Irradiation with D Atoms

In [9], adsorption of thermal (2000 K) D atoms on (0001) surfaces of various highly oriented pyrolytic graphite (HOPG) was studied under ultra-high vacuum conditions with thermal desorption spectroscopy (TDS, see Figures 6–8). Disc shaped HOPG samples with a diameter of 9 mm and thickness of 1–2 mm, consisting of well aligned crystallites about 5 μm in diameter, were used.

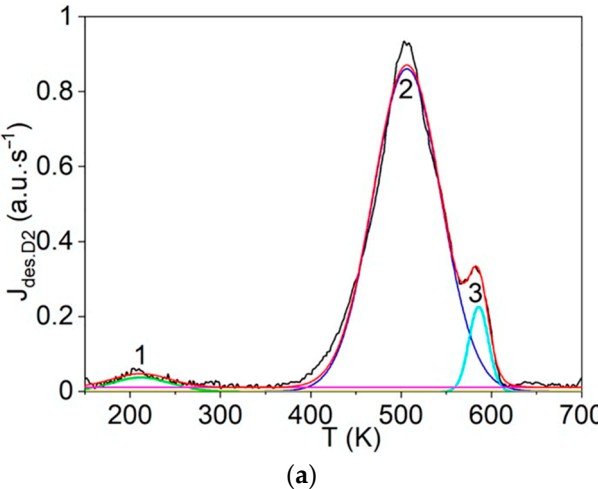
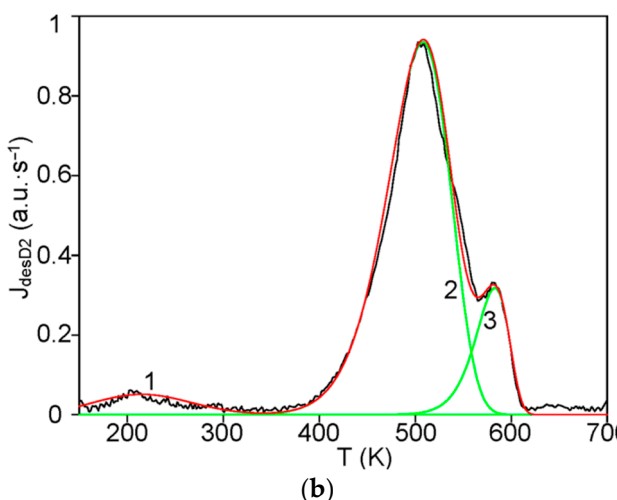

(**a**)  (**b**)

**Figure 6.** Deconvolution of the thermal desorption spectrum (from Figure 8c in [9]) measured after admitting D atoms (exposure 12 ML, 1 ML = 3.8 × 10$^{15}$ cm$^{-2}$) to clean HOPG surfaces at 150 K, as follows: (**a**) By three Gaussians (peaks #1–3), with the help of the methodology [20]. (**b**) By three non-Gaussians, with the help of the numerical simulation [20,21], in the approximation of the first-order reactions.

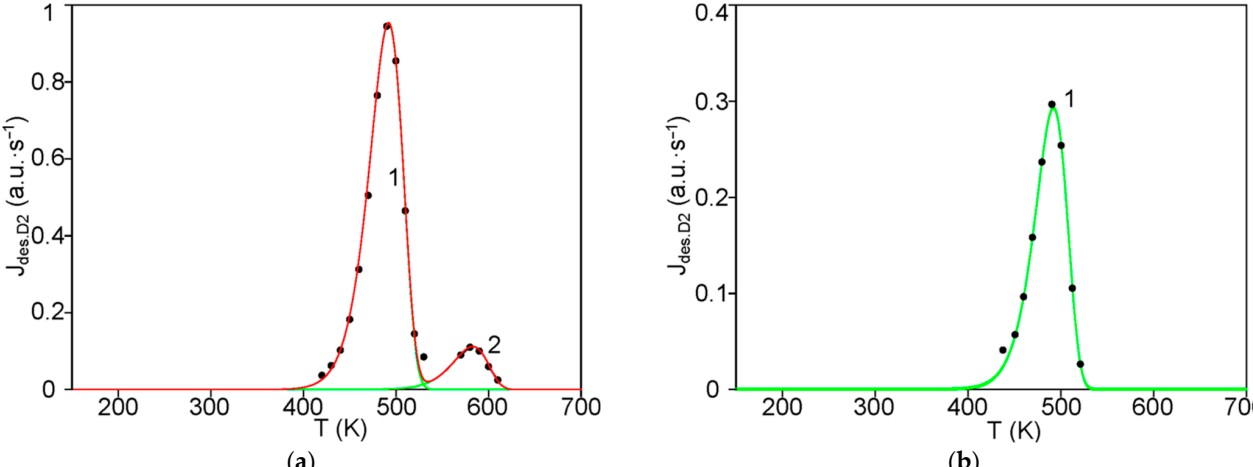

**Figure 7.** Deconvolution, with the help of the numerical simulation [20,21], of thermal desorption spectrum measured after admitting D atoms to clean HOPG surfaces at 150 K, in the approximation of the first-order reactions, as follows: (**a**) By two non-gaussians for exposure 0.4 ML (Figure 8b in [9]). (**b**) By one non-Gaussian for exposure 0.05 ML (Figure 8a in [9]); 1 ML = $3.8 \times 10^{15}$ cm$^{-2}$.

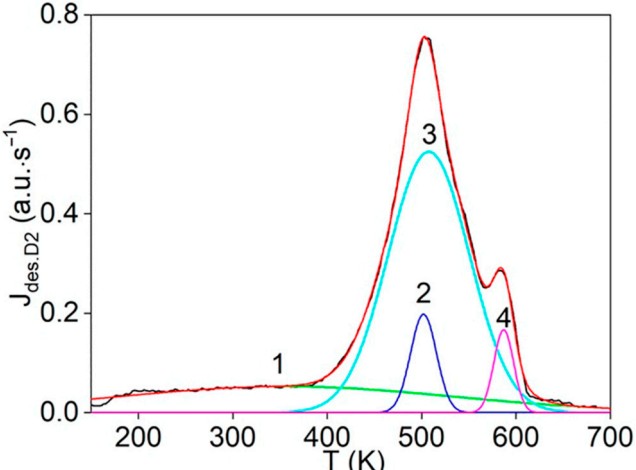

**Figure 8.** Deconvolution by four Gaussians (peaks #1–4) of the thermal desorption spectrum ((from Figure 6 (HOPG-ZYN+2300 K) in [9])) measured after admitting D atoms (exposure 12 ML, 1 ML = $3.8 \times 10^{15}$ cm$^{-2}$) to clean HOPG surfaces at 150 K.

In order to extract kinetic data from the TD spectra shown in Figure 8, they were analyzed in [9] with the assumption that four separate first-order desorption processes contributed peaks ## 1–4. The activation energies ($Q_{\#1(1\_ord.)[9]}$ = 106 kJ mol$^{-1}$, $Q_{\#2(1\_ord.)[9]}$ = 111 kJ mol$^{-1}$, $Q_{\#3(1\_ord.)[9]}$ = 116 kJ mol$^{-1}$, $Q_{\#4(1\_ord.)[9]}$ = 159 kJ mol$^{-1}$) for these four peaks were obtained in [9] from a leading edge analysis [25] of measured spectra. The appropriate frequency factors ($K_{0\#1(1\_ord.)[9]} \approx 10^{10}$ s$^{-1}$, $K_{0\#2(1\_ord.)[9]} \approx 10^{10}$ s$^{-1}$, $K_{0\#3(1\_ord.)[9]} \approx 10^{10}$ s$^{-1}$, $K_{0\#4(1\_ord.)[9]} \approx 10^{13}$ s$^{-1}$) were found in [9] by adjusting the calculated desorption peak temperature to the measured value. For the TD spectra, shown in Figures 6 and 7, the activation energies for the main peak #1m were obtained in [9] from a leading edge analysis [25] with respect to D coverage ($Q_{\#1m\_0.05(1\_ord.)[9]}$ = 96–116 kJ mol$^{-1}$ (for dose 0.05 ML), $Q_{\#1m\_0.4(1\_ord.)[9]}$ = 96 kJ mol$^{-1}$ (for dose 0.4 ML), $Q_{\#1m\_12(1\_ord.)[9]}$ = 48 kJ mol$^{-1}$ (for a dose of 12 ML)).

These results [9] on $Q$ and $K_0$ quantities are satisfactory (within the errors) consistent with our results presented below in this Section (see Figures 6–8 and Tables A7–A10).

The results of the numerical simulation [20,21] of the three non-Gaussian peaks in Figure 6b (Table A8) are satisfactorily consistent (within the errors for the $Q$ and ln $K_0$ quantities) with the results of processing of the three Gaussians (#1, #2, #3) in Figure 6a (Table A7).

### 3.12. Results of Processing the TDS Data for Isotropic Graphite after Irradiation with Atomic Hydrogen in Plasma

In [12], results of a thermal desorption study of states of deuterium in carbon materials and nanomaterials, including isotropic fine-grained graphite MPG-8, after plasma exposure have been presented. The experiments [12] were performed at the National Research Nuclear University MEPHI. The samples [12] were imbedded under the floating potential in deuterium plasma of abnormal glow discharge at the pressure of 1 mbar for 30–90 min and at temperatures of 510–750 K.

Below, see Figure 9a,b and Table A11, one of TPD spectra [12] for isotropic graphite MPG-8 is under consideration.

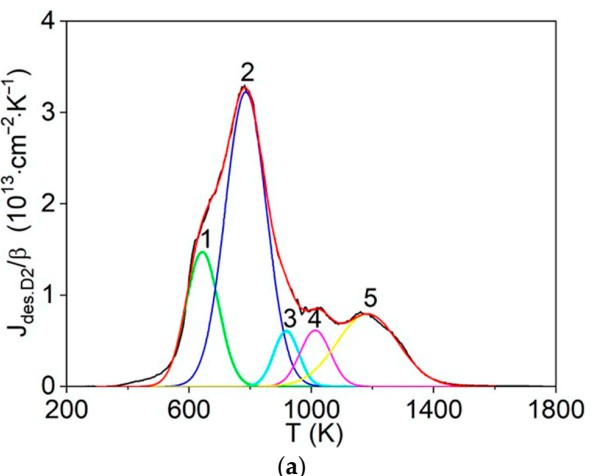

(**a**)

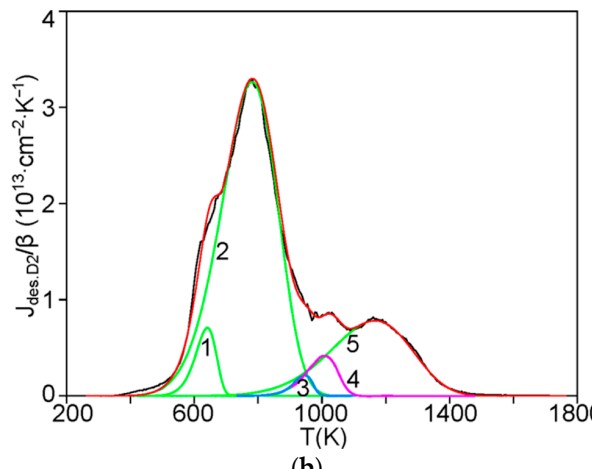

(**b**)

**Figure 9.** Deconvolution of the thermal desorption spectrum (the heating rate β = 2 K s$^{-1}$ [12]) of deuterium for isotropic graphite after plasma exposure (90 min, 510 K), as follows: (**a**) By five Gaussians (peaks #1–5). (**b**) By five non-Gaussians, with the help of the numerical simulation [20,21], in the approximation of the first-order reactions.

The results of the numerical simulation of the non-Gaussian peaks in Figure 9b are satisfactorily consistent (within the errors, up to 15% for the Q and ln $K_0$ quantities) with the results of processing of the main Gaussians (#2 and #5) in Figure 9a (Table A11).

The obtained $Q$ and $K_0$ quantities are close (within the errors) to the related quantities for some desorption processes (peaks) considered and interpreted in the previous sections.

### 3.13. Comparison with the Related Results of Processing the TDS Data for Nanoscale Carbon Structures

In [20], results of a similar thermal desorption study of states of hydrogen in nanoscale carbon structures (carbon nanosheets [5], graphitized nanodiamonds, hydrogenated single and few-layer epitaxial graphene [6]) have been presented. Comparison of the study results shows that in the micro- and nanoscale carbon structures, there are both close and significantly different states of hydrogen.

## 4. Conclusions

The efficient methodology of the detailed analysis of thermal desorption spectra of hydrogen in carbon materials and nanomaterials has been further developed and applied for a number of experimental data for graphite structures subjected to irradiation with atomic hydrogen. The methodology is based on a definite approximation by Gaussian peaks

of TDS and a definite processing of the Gaussians and the corresponding non-Gaussians, in the approximation of the first-order and the second-order reactions.

In such a way, the activation energies and pre-exponential factors of the rate constants of desorption processes, corresponding to the peaks (Gaussians and non-Gaussians) with different temperatures of the maximum desorption rate, have been defined, along with revealing the characteristics and possible models of different chemisorption and physisorption states of hydrogen in graphite materials after irradiation with atomic hydrogen, and particularly, the role of the subgrain boundaries, grain boundaries and free surfaces in the desorption processes.

This technique is not less informative, but much less time-consuming in experimental terms compared to the generally accepted Kissinger method, which is usually applied only for the first-order desorption processes, demands the use of several heating rates, and has strict limits of applicability. As is shown in this study, the Kissinger method can be also applied for the second-order desorption processes (reactions).

The main research finding of this work is further refinement and/or disclosure of poorly studied characteristics and physics of various states of hydrogen in microscale graphite structures after irradiation with atomic hydrogen, and comparison with the related results for nanoscale carbon structures with different hydrogen content.

**Author Contributions:** Conceptualization, Y.S.N., E.A.D., A.Ö. and S.Y.D.; methodology, Y.S.N., E.A.D. and E.K.K.; software, E.K.K., E.A.D., N.A.S. and A.O.C.; validation, Y.S.N., S.Y.D., E.A.D. and A.O.C.; formal analysis, A.Ö., S.Y.D. and N.A.S.; investigation, E.A.D., E.K.K., N.A.S. and A.O.C.; resources, Y.S.N. and E.A.D.; data curation, E.A.D., N.A.S., A.O.C. and E.K.K.; writing—original draft preparation, Y.S.N.; writing—review and editing, A.Ö., E.A.D. and A.O.C.; visualization, S.Y.D. and A.Ö.; supervision, Y.S.N.; project administration, Y.S.N.; funding acquisition, Y.S.N. All authors have read and agreed to the published version of the manuscript.

**Funding:** This work was financially supported by the Russian foundation for basic research (Project # 18-29-19149 mk).

**Institutional Review Board Statement:** Not applicable.

**Informed Consent Statement:** Not applicable.

**Data Availability Statement:** All data analyzed during this study are included in this published article.

**Conflicts of Interest:** The authors declare no conflict of interest.

## Appendix A

**Table A1.** Results of processing of the six Gaussians (peaks ##1–6 in Figure 1a) in the approximation of the first- and second-order reactions. The quantity $\gamma$ is the peak fraction of the spectrum; the quantity $C_0 = \gamma \cdot (H/C)_{\Sigma vol.}$ is the initial atomic fraction of hydrogen corresponding to the peak; the quantity $(H/C)_{\Sigma vol.} \sim 2.5 \times 10^{-7}$ is the total initial atomic fraction of hydrogen.

| Peak # | $T_{max}$, K | Order Reactions | $Q$, kJ mol$^{-1}$ | $K_0$, s$^{-1}$ | $K(T_{max})$, s$^{-1}$ | $Q^*$, kJ mol$^{-1}$ | $\gamma$ | $C_0$ |
|--------|--------------|-----------------|--------------------|------------------|-------------------------|----------------------|----------|--------|
| 1 | 1001 | First | 163 | $1.5 \times 10^8$ | 0.47 | 162 | 0.02 | $5.0 \times 10^{-9}$ |
|   |      | Second | 326 | $1.0 \times 10^{17}$ | 0.97 | 325 |  |  |
| 2 | 1145 | First | 201 | $7.3 \times 10^8$ | 0.49 | 201 | 0.25 | $6.2 \times 10^{-8}$ |
|   |      | Second | 402 | $2.2 \times 10^{18}$ | 1.00 | 401 |  |  |
| 3 | 1226 | First | 373 | $5.7 \times 10^{15}$ | 0.73 | 371 | 0.04 | $1.0 \times 10^{-8}$ |
|   |      | Second | 745 | $8.6 \times 10^{31}$ | 1.50 | 742 |  |  |
| 4 | 1300 | First | 227 | $5.3 \times 10^8$ | 0.40 | 225 | 0.14 | $3.5 \times 10^{-8}$ |
|   |      | Second | 452 | $1.2 \times 10^{18}$ | 0.82 | 451 |  |  |
| 5 | 1389 | First | 441 | $2.6 \times 10^{16}$ | 0.68 | 437 | 0.04 | $1.0 \times 10^{-8}$ |
|   |      | Second | 876 | $1.2 \times 10^{33}$ | 1.40 | 874 |  |  |
| 6 | 1538 | First | 189 | $6.3 \times 10^5$ | 0.24 | 188 | 0.51 | $1.3 \times 10^{-7}$ |
|   |      | Second | 377 | $3.0 \times 10^{12}$ | 0.47 | 376 |  |  |

**Table A2.** Results of processing of the six non-Gaussians in the approximation of the first- and second-order reactions.

| Peak # | $T_{max}$, K | Order Reactions | $Q$, kJmol$^{-1}$ | $K_0$, s$^{-1}$ | $K(T_{max})$, s$^{-1}$ | $\gamma$ | $\theta(T_{max})$ | $C_0$ |
|---|---|---|---|---|---|---|---|---|
| 1 | 950 | first | 162 | $4.4 \times 10^8$ | 0.54 | 0.01 | 0.41 | $2.5 \times 10^{-9}$ |
| | 980 | second | 326 | $2.3 \times 10^{17}$ | 0.97 | 0.02 | 0.53 | $5.0 \times 10^{-9}$ |
| 2 | 1122 | first | 230 | $2.8 \times 10^{10}$ | 0.55 | 0.22 | 0.40 | $5.5 \times 10^{-8}$ |
| | 1129 | second | 340 | $4.1 \times 10^{15}$ | 0.76 | 0.26 | 0.53 | $6.5 \times 10^{-8}$ |
| 3 | 1198 | first | 371 | $1.2 \times 10^{16}$ | 0.78 | 0.05 | 0.39 | $1.3 \times 10^{-8}$ |
| | 1198 | second | 745 | $2.0 \times 10^{32}$ | 1.49 | 0.03 | 0.52 | $7.5 \times 10^{-9}$ |
| 4 | 1259 | first | 225 | $9.2 \times 10^8$ | 0.43 | 0.13 | 0.40 | $3.2 \times 10^{-8}$ |
| | 1259 | second | 385 | $3.0 \times 10^{15}$ | 0.67 | 0.13 | 0.53 | $3.3 \times 10^{-8}$ |
| 5 | 1360 | first | 441 | $6.2 \times 10^{16}$ | 0.72 | 0.04 | 0.39 | $1.0 \times 10^{-8}$ |
| | 1360 | second | 807 | $7.0 \times 10^{30}$ | 1.26 | 0.03 | 0.52 | $7.5 \times 10^{-9}$ |
| 6 | 1526 | first | 212 | $4.9 \times 10^6$ | 0.27 | 0.55 | 0.41 | $1.4 \times 10^{-7}$ |
| | 1526 | second | 320 | $3.9 \times 10^{10}$ | 0.39 | 0.53 | 0.54 | $1.3 \times 10^{-7}$ |

**Table A3.** Results of processing of the two Gaussians (peaks ## 1,2 in Figure 2a) in the approximation of the first- and second-order reactions.

| Peak # | $T_{max}$, K | Order Reactions | $Q$, kJ mol$^{-1}$ | $K_0$, s$^{-1}$ | $K(T_{max})$, s$^{-1}$ | $Q^*$, kJ mol$^{-1}$ | $\gamma$ |
|---|---|---|---|---|---|---|---|
| 1 | 469 | first | 64.6 | $5.5 \times 10^5$ | $3.5 \times 10^{-2}$ | 64.5 | 0.55 |
| | | second | 130 | $2.1 \times 10^{13}$ | $7.1 \times 10^{-2}$ | 129 | |
| 2 | 584 | first | 54.6 | $1.5 \times 10^3$ | $1.9 \times 10^{-2}$ | 54.5 | 0.45 |
| | | second | 110 | $2.5 \times 10^8$ | $3.9 \times 10^{-2}$ | 110 | |

**Table A4.** Results of processing of the Gaussians (peaks in Figure 3) in the approximation of the first- and second-order reactions.

| Peak # | $T_{max}$, K | Order Reactions | $Q$, kJ mol$^{-1}$ | $K_0$, s$^{-1}$ | $K(T_{max})$, s$^{-1}$ | $Q^*$, kJ mol$^{-1}$ | $\gamma$ | $C_0$ |
|---|---|---|---|---|---|---|---|---|
| | | | | peaks ##1–4 in Figure 3a | | | | |
| 1 | 353 | first | 6.2 | $4.9 \times 10^{-2}$ | $5.9 \times 10^{-3}$ | 6.1 | 0.24 | $1.2 \times 10^{-8}$ |
| | | second | 13 | $9.3 \times 10^{-1}$ | 0.11 | 11.5 | | |
| 2 | 436 | first | 60.8 | $7.4 \times 10^5$ | $3.8 \times 10^{-2}$ | 60.3 | 0.04 | $2.1 \times 10^{-9}$ |
| | | second | 120 | $2.2 \times 10^{13}$ | $7.6 \times 10^{-2}$ | 120 | | |
| 3 | 508 | first | 52.0 | $5.4 \times 10^3$ | $2.4 \times 10^{-2}$ | 51.9 | 0.67 | $3.2 \times 10^{-8}$ |
| | | second | 104 | $2.5 \times 10^9$ | $4.8 \times 10^{-2}$ | 104 | | |
| 4 | 584 | first | 208 | $3.0 \times 10^{17}$ | $7.3 \times 10^{-2}$ | 207 | 0.05 | $2.2 \times 10^{-9}$ |
| | | second | 415 | $2.4 \times 10^{36}$ | 0.15 | 415 | | |
| | | | | peaks ##1–4 in Figure 3b | | | | |
| 1 | 361 | first | 6.3 | $4.7 \times 10^{-2}$ | $5.7 \times 10^{-3}$ | 6.2 | 0.19 | $6.9 \times 10^{-9}$ |
| | | second | 13.1 | $8.7 \times 10^{-1}$ | $1.1 \times 10^{-2}$ | 11.7 | | |
| 2 | 497 | first | 128 | $1.7 \times 10^{12}$ | $6.1 \times 10^{-2}$ | 126 | 0.11 | $4.2 \times 10^{-9}$ |
| | | second | 251 | $2.7 \times 10^{25}$ | 0.12 | 251 | | |
| 3 | 500 | first | 42.3 | $5.4 \times 10^2$ | $2.0 \times 10^{-2}$ | 42.1 | 0.66 | $2.4 \times 10^{-8}$ |
| | | second | 84.4 | $2.8 \times 10^7$ | $4.0 \times 10^{-2}$ | 84.2 | | |
| 4 | 581 | first | 212 | $8.6 \times 10^{17}$ | $7.6 \times 10^{-2}$ | 213 | 0.04 | $1.6 \times 10^{-9}$ |
| | | second | 430 | $7.0 \times 10^{37}$ | 0.15 | 427 | | |

**Table A4.** *Cont.*

| Peak # | $T_{max}$, K | Order Reactions | $Q$, kJ mol$^{-1}$ | $K_0$, s$^{-1}$ | $K(T_{max})$, s$^{-1}$ | $Q^*$, kJ mol$^{-1}$ | $\gamma$ | $C_0$ |
|---|---|---|---|---|---|---|---|---|
| | | | | peaks ##1–3 in Figure 3c | | | | |
| 1 | 354 | first | 6.5 | $5.6 \times 10^{-2}$ | $6.2 \times 10^{-3}$ | 6.4 | 0.18 | $4.5 \times 10^{-9}$ |
| | | second | 13.4 | 1.2 | $1.2 \times 10^{-2}$ | 12.2 | | |
| 2 | 491 | first | 73.0 | $2.2 \times 10^6$ | $3.7 \times 10^{-2}$ | 73.1 | 0.67 | $1.6 \times 10^{-8}$ |
| | | second | 147 | $3.5 \times 10^{14}$ | $7.3 \times 10^{-2}$ | 146 | | |
| 3 | 567 | first | 90.8 | $8.0 \times 10^6$ | $3.4 \times 10^{-2}$ | 90.4 | 0.15 | $3.6 \times 10^{-9}$ |
| | | second | 181 | $3.7 \times 10^{15}$ | $6.8 \times 10^{-2}$ | 181 | | |
| | | | | peaks ##1–5 in Figure 3d | | | | |
| 1 | 457 | first | 12.6 | $2.0 \times 10^{-1}$ | $7.3 \times 10^{-3}$ | 12.6 | 0.16 | $2.3 \times 10^{-9}$ |
| | | second | 24.9 | $1.1 \times 10^1$ | $1.5 \times 10^{-2}$ | 25.6 | | |
| 2 | 477 | first | 76.2 | $9.0 \times 10^6$ | $4.0 \times 10^{-2}$ | 75.8 | 0.37 | $5.3 \times 10^{-9}$ |
| | | second | 152 | $4.0 \times 10^{15}$ | $8.0 \times 10^{-2}$ | 152 | | |
| 3 | 493 | first | 132 | $6.0 \times 10^{12}$ | $6.4 \times 10^{-2}$ | 130 | 0.33 | $4.7 \times 10^{-9}$ |
| | | second | 260 | $5.4 \times 10^{26}$ | 0.13 | 261 | | |
| 4 | 540 | first | 157 | $9.2 \times 10^{13}$ | $6.4 \times 10^{-2}$ | 156 | 0.05 | $7.0 \times 10^{-10}$ |
| | | second | 314 | $3.1 \times 10^{29}$ | 0.13 | 313 | | |
| 5 | 577 | first | 162 | $2.9 \times 10^{13}$ | $5.8 \times 10^{-2}$ | 161 | 0.09 | $1.3 \times 10^{-9}$ |
| | | second | 322 | $1.8 \times 10^{28}$ | 0.12 | 322 | | |
| | | | | peaks ##1–3 in Figure 3e | | | | |
| 1 | 451 | first | 13.2 | $2.6 \times 10^{-1}$ | $7.8 \times 10^{-3}$ | 13.2 | 0.25 | $1.8 \times 10^{-9}$ |
| | | second | 26.1 | $1.7 \times 10^1$ | $1.6 \times 10^{-2}$ | 26.5 | | |
| 2 | 485 | first | 99.6 | $2.7 \times 10^9$ | $5.1 \times 10^{-2}$ | 100 | 0.65 | $4.7 \times 10^{-9}$ |
| | | second | 202 | $6.5 \times 10^{20}$ | 0.10 | 200 | | |
| 3 | 575 | first | 135 | $9.6 \times 10^{10}$ | $4.9 \times 10^{-2}$ | 134 | 0.10 | $7.1 \times 10^{-10}$ |
| | | second | 268 | $2.0 \times 10^{23}$ | $9.7 \times 10^{-2}$ | 268 | | |
| | | | | peaks ##1–3 in Figure 3f | | | | |
| 1 | 411 | first | 8.4 | $7.0 \times 10^{-2}$ | $6.0 \times 10^{-3}$ | 8.4 | 0.33 | $1.2 \times 10^{-9}$ |
| | | second | 16.8 | 1.7 | $1.2 \times 10^{-2}$ | 16.7 | | |
| 2 | 484 | first | 104.5 | $1.0 \times 10^{10}$ | $5.4 \times 10^{-2}$ | 104.6 | 0.61 | $2.2 \times 10^{-9}$ |
| | | second | 211 | $6.0 \times 10^{21}$ | 0.11 | 209 | | |
| 3 | 576 | first | 160 | $1.7 \times 10^{13}$ | $5.7 \times 10^{-2}$ | 158 | 0.06 | $2.3 \times 10^{-10}$ |
| | | second | 317 | $6.4 \times 10^{27}$ | 0.12 | 317 | | |

**Table A5.** Results of processing of the non-Gaussians (peaks ## 1–4 in Figure 4) in the approximation of the first-order reactions.

| Peak # | $T_{max}$, K | $Q$, kJ mol$^{-1}$ | $K_0$, s$^{-1}$ | $K(T_{max})$, s$^{-1}$ | $\gamma$ | $\theta_{max}$ |
|---|---|---|---|---|---|---|
| 1 | 457 | 12.6 | $2.0 \times 10^{-1}$ | $7.3 \times 10^{-3}$ | 0.15 | 0.52 |
| 2 | 490 | 120 | $3.9 \times 10^{11}$ | $6.0 \times 10^{-2}$ | 0.68 | 0.39 |
| 3 | 532 | 148 | $2.1 \times 10^{13}$ | $6.3 \times 10^{-2}$ | 0.05 | 0.39 |
| 4 | 578 | 161 | $2.0 \times 10^{13}$ | $5.8 \times 10^{-2}$ | 0.12 | 0.39 |

**Table A6.** Results of processing of the four Gaussians (peaks ##1–4 in Figure 5) in the approximation of the first- and second-order reactions.

| Peak # | $T_{max}$, K | Order Reactions | $Q$, kJ mol$^{-1}$ | $K_0$, s$^{-1}$ | $K(T_{max})$, s$^{-1}$ | $Q^*$, kJ mol$^{-1}$ | $\gamma$ | $C_0$ |
|---|---|---|---|---|---|---|---|---|
| 1 | 420 | first | 21.5 | 7.0 | $1.5 \times 10^{-2}$ | 21.4 | 0.25 | $9.9 \times 10^{-9}$ |
| | | second | 42.9 | $6.5 \times 10^3$ | $2.9 \times 10^{-2}$ | 42.9 | | |
| 2 | 472 | first | 49.0 | $6.9 \times 10^3$ | $2.6 \times 10^{-2}$ | 48.7 | 0.62 | $2.5 \times 10^{-8}$ |
| | | second | 97.4 | $3.2 \times 10^9$ | $5.2 \times 10^{-2}$ | 97.4 | | |
| 3 | 507.5 | first | 160 | $2.2 \times 10^{15}$ | $7.4 \times 10^{-2}$ | 159 | 0.06 | $2.4 \times 10^{-9}$ |
| | | second | 319 | $1.1 \times 10^{32}$ | $1.5 \times 10^{-1}$ | 319 | | |
| 4 | 563.5 | first | 170 | $4.0 \times 10^{14}$ | $6.4 \times 10^{-2}$ | 170 | 0.07 | $2.6 \times 10^{-9}$ |
| | | second | 341 | $5.0 \times 10^{30}$ | 0.13 | 340 | | |

**Table A7.** Results of processing of the three Gaussians (peaks ##1–3 in Figure 6a) in the approximation of the first- and second-order reactions.

| Peak # | $T_{max}$, K | Order Reactions | $Q$, kJ mol$^{-1}$ | $K_0$, s$^{-1}$ | $K(T_{max})$, s$^{-1}$ | $Q^*$, kJ mol$^{-1}$ | $\gamma$ |
|---|---|---|---|---|---|---|---|
| 1 | 212 | first | 9.2 | 4.5 | $2.5 \times 10^{-2}$ | 9.2 | 0.03 |
| | | second | 18.9 | $2.3 \times 10^3$ | $4.8 \times 10^{-2}$ | 18.0 | |
| 2 | 506 | first | 45.6 | $1.1 \times 10^3$ | $2.1 \times 10^{-2}$ | 45.4 | 0.90 |
| | | second | 91.2 | $1.1 \times 10^8$ | $4.3 \times 10^{-2}$ | 90.9 | |
| 3 | 586 | first | 212 | $6.2 \times 10^{17}$ | $7.4 \times 10^{-2}$ | 212 | 0.07 |
| | | second | 426 | $1.5 \times 10^{37}$ | 0.15 | 424 | |

**Table A8.** Results of processing of the three peaks (non-Gaussians) in Figure 6b in the approximation of the first-order reactions.

| Peak #. | $T_{max}$, K | $Q$, kJ mol$^{-1}$ | $K_0$, s$^{-1}$ | $K(T_{max})$, s$^{-1}$ | $\gamma$ | $\theta_{max}$ |
|---|---|---|---|---|---|---|
| 1 | 215 | 6.0 | $4.5 \times 10^{-1}$ | $1.6 \times 10^{-2}$ | 0.06 | 0.56 |
| 2 | 508 | 62.0 | $6.8 \times 10^4$ | $2.9 \times 10^{-2}$ | 0.79 | 0.41 |
| 3 | 583 | 160 | $1.2 \times 10^{13}$ | $5.7 \times 10^{-2}$ | 0.15 | 0.39 |

**Table A9.** Results of processing of the two peaks (non-Gaussians) in Figure 7a and the one peak (non-Gaussian) in Figure 7b in the approximation of the first-order reactions.

| Peak # | $T_{max}$, K | $Q$, kJ mol$^{-1}$ | $K_0$, s$^{-1}$ | $K(T_{max})$, s$^{-1}$ | $\gamma$ | $\theta_{max}$ |
|---|---|---|---|---|---|---|
| | | the two peaks (non-Gaussians) in Figure 7a | | | | |
| 1 | 492 | 108 | $1.6 \times 10^{10}$ | $5.4 \times 10^{-2}$ | 0.89 | 0.40 |
| 2 | 583 | 145 | $5.0 \times 10^{11}$ | $5.1 \times 10^{-2}$ | 0.11 | 0.39 |
| | | the peak (non-Gaussian) in Figure 7b | | | | |
| 1 | 492 | 113 | $5.6 \times 10^{10}$ | $5.6 \times 10^{-2}$ | 1.0 | 0.40 |

**Table A10.** Results of processing of the four Gaussians (peaks ##1–4 in Figure 8) in the approximation of the first- and second-order reactions.

| Peak # | $T_{max}$, K | Order Reactions | $Q$, kJ mol$^{-1}$ | $K_0$, s$^{-1}$ | $K(T_{max})$, s$^{-1}$ | $Q^*$, kJ mol$^{-1}$ | $\gamma$ |
|---|---|---|---|---|---|---|---|
| 1 | 356 | first | 5.1 | $3.0 \times 10^{-2}$ | $4.7 \times 10^{-3}$ | 5.0 | 0.23 |
| | | second | 10.6 | $3.3 \times 10^{-1}$ | $9.0 \times 10^{-3}$ | 9.5 | |
| 2 | 502 | first | 122 | $3.2 \times 10^{11}$ | $5.8 \times 10^{-2}$ | 122 | 0.08 |
| | | second | 244 | $3.0 \times 10^{24}$ | 0.12 | 243 | |
| 3 | 508 | first | 39.8 | $2.3 \times 10^{2}$ | $1.9 \times 10^{-2}$ | 39.7 | 0.64 |
| | | second | 80.0 | $6.3 \times 10^{6}$ | $3.7 \times 10^{-2}$ | 79.5 | |
| 4 | 587 | first | 219 | $2.4 \times 10^{18}$ | $7.6 \times 10^{-2}$ | 218 | 0.05 |
| | | second | 437 | $1.3 \times 10^{38}$ | 0.15 | 436 | |

**Table A11.** Results of processing of the five Gaussians (peaks ##1–5 in Figure 9a) in the approximation of the first- and second-order reaction.

| Peak # | $T_{max}$, K | Order Reactions | $Q$, kJmol$^{-1}$ | $K_0$, s$^{-1}$ | $K(T_{max})$, s$^{-1}$ | $Q^*$, kJmol$^{-1}$ | $\gamma$ | $C_0$ |
|---|---|---|---|---|---|---|---|---|
| 1 | 643 | first | 51.5 | $4.5 \times 10^{2}$ | $3.0 \times 10^{-2}$ | 51.2 | 0.18 | $2.0 \times 10^{-6}$ |
| | | second | 103 | $2.5 \times 10^{7}$ | $6.0 \times 10^{-2}$ | 103 | | |
| 2 | 786 | first | 61.2 | $2.8 \times 10^{2}$ | $2.4 \times 10^{-2}$ | 61.0 | 0.50 | $5.4 \times 10^{-6}$ |
| | | second | 122 | $5.6 \times 10^{6}$ | $4.8 \times 10^{-2}$ | 122 | | |
| 3 | 918 | first | 138 | $2.7 \times 10^{6}$ | $3.9 \times 10^{-2}$ | 137 | 0.06 | $6.1 \times 10^{-7}$ |
| | | second | 276 | $9.3 \times 10^{12}$ | $7.9 \times 10^{-2}$ | 275 | | |
| 4 | 1013 | first | 138 | $4.2 \times 10^{5}$ | $3.2 \times 10^{-2}$ | 137 | 0.07 | $7.6 \times 10^{-7}$ |
| | | second | 276 | $1.3 \times 10^{17}$ | $6.5 \times 10^{-2}$ | 275 | | |
| 5 | 1183 | first | 93.4 | $2.1 \times 10^{2}$ | $1.6 \times 10^{-2}$ | 93.0 | 0.19 | $2.0 \times 10^{-6}$ |
| | | second | 186 | $1.9 \times 10^{6}$ | $3.2 \times 10^{-2}$ | 186 | | |

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
