# Peer review of "Revealing Hydrogen States in Carbon Structures by Analyzing the Thermal Desorption Spectra"

_carbon_

Round 1
Reviewer 1 Report
Thank you for inviting me to review the manuscript titled “Revealing hydrogen states in carbon structures by analyzing the thermal desorption spectra”. The article represents the application of thermal desorption spectra (TDS) to analyze hydrogen in carbon structures of micro- and nano- scale. In particular, it defines the state of hydrogen in microscale graphite structures. I believe that some points should be considered before acceptance for publication:
- A comparison with some existing methods regarding hydrogen for various carbon materials and nano- materials should be presented.
- A real case to support the proposed method should be given.
- The authors should unveil the major gaps within the existing knowledge of the proposed hydrogen analysis method.
- The abstract and conclusion sections should be improved to show the main research findings.
- Do not repeat the words in the title to the “keywords”.
- The last paragraph of Introduction should include the study objectives/procedures in brief.
- The English in the present manuscript requires improvement. Please carefully proof-read spell check to eliminate grammatical errors.
Reviewer 2 Report
The paper presents research on revealing hydrogen states in carbon structures by analyzing the thermal desorption spectra. The presentation of methods and scientific results in the current form is satisfactory for publication in the C Journal of Carbon Research. The minor and significant drawbacks to be addressed can be specified as follows:
- It looks pretty strange to use * to number the equations – i.e. Eq. (1*) instead of Eq. (3). See also Eq. (3*).
- 1, figure captions. Problem with subscripts and superscripts, i.e. (i) Tirr (ii) cm-2. Please, check all the paper.
- 1. Column K(Tmax). For example, replace 4.7*10^-1 by 0.47. Replace 1.0*10^0 by 1.00. See also Tabs. 2 and 3 – the notation for a column with the values of the same quantity.
- Why are Tables 2 and 3 not combined? In Table 1, the results for the first- and second-order reactions are listed together.
- 6. 3.2. Interpretation of peak #2 in Figs. 1a,b. The paper is quite hard to read. There are many references to earlier articles by the authors - it does not make it easier to read and understand what the authors write about.
- Lines 249 and 250 – (…)for peak #4 in Figs. 1a,b (…). Page. 6. 3.2. Interpretation of peak #2 in Figs. 1a,b. ---> Page. 6. 3.2. Interpretation of peak #2 and #4 in Figs. 1a,b.
- Line 478. #3 and #2 ---> #2 and #3.
- Conclusions. After reading the article, I do not know what the result of this research shows. What is the weight fraction of hydrogen in the individual materials? I think it would be interesting to perform measurements on a series of carbon materials that differ in the degree of hydrogenation. The use of such a series would answer the credibility of the authors' deliberations and the continuity, or lack thereof, of changes.
- At least 10 self-citations ([6], [11], [15-20], [22], and [25]) on 25 references 40% (not acceptable).
Reviewer 3 Report
The paper presents a technique to analyze the thermal desorption spectra of hydrogen in carbon structures of micro- and nanoscale. Such a technique allows the determination of the activation energies and the pre-exponential factors of the rate constants of the hydrogen desorption processes and is applied to several graphite materials irradiated with atomic hydrogen.
The study contains useful information that can help to understand the behavior of hydrogen in carbon-based materials and nanomaterials.
The paper could be suitable for publication after a revision:
1. “ Carbon materials and nanomaterials, especially those based on graphene, are widely 35 used as energetic materials associated with electrochemical conversion and energy stor-36 age in fuel cells, supercapacitors, and lithium-ion batteries [1].” Here is an additional reference that could be added to support this statement: https://doi.org/10.1016/j.jallcom.2020.158497
2. “Then, the quantity Q* (related to the quantity Q) was evaluated, by using the corre-88 sponding expression for the first-order reactions (Eqn. (23) in [15]) and the values of Tmax 89 and K(Tmax) for the Gaussian under consideration.” To increase the self-consistency of the paper, I suggest reporting the mentioned eq. 23 of ref [15]. This comment applies also to other parts where there is an explicit reference to other papers that cannot be easily accessed.
3. Section 3.2. The authors mention “the subgrain size for graphite materials”. They should clarify what they mean by subgrain and grain in the graphite materials.
4. As presented the paper is rather long and heavy to read. The authors might optionally consider splitting the content of their study into a shorter paper and an accompanying supporting information file.
Round 2
Reviewer 1 Report
The authors' responses are satisfactory
Reviewer 2 Report
The authors revised their manuscript well. The revised version can be accepted for publication.